# Symptom severity trajectories and distresses in patients undergoing video-assisted thoracoscopic lung resection from surgery to the first post-discharge clinic visit

**Tomohito Saito** [1]*, **Anna Hamakawa**[2], **Hideto Takahashi**[3], **Yukari Muto**[2], **Miku Mouri**[2], **Makie Nakashima**[2], **Natsumi Maru**[1], **Takahiro Utsumi**[1], **Hiroshi Matsui**[1], **Yohei Taniguchi**[1], **Haruaki Hino**[1], **Emi Hayashi**[2], **Tomohiro Murakawa**[1], on behalf of the SMILE-001 investigators[¶]

**1** Department of Thoracic Surgery, Kansai Medical University, Hirakata, Osaka, Japan, **2** Nursing Department, Kansai Medical University Hospital, Hirakata, Osaka, Japan, **3** National Institute of Public Health, Wako, Saitama, Japan

¶ A complete list of investigators in the Seamless Monitoring of Illness after Lung surgery for Enhanced support (SMILE)-001 study is provided in Acknowledgments section.
* saitotom@hirakata.kmu.ac.jp

**Data Availability Statement:** Data are available from the Kansai Medical University Research Data Online Storage (http://rdos.tnoc.kmu.ac.jp/) for

## Abstract

This study aimed to characterize patients' symptom severity trajectories and distresses from video-assisted thoracoscopic lung resection to the first post-discharge clinic visit. Seventy-five patients undergoing thoracoscopic lung resection for diagnosed or suspected pulmonary malignancy prospectively recorded daily symptom severity on a 0–10 numeric scale using the MD Anderson Symptom Inventory until the first post-discharge clinic visit. The causes of postoperative distresses were surveyed, and symptom severity trajectories were analyzed using joinpoint regression. A rebound was defined as a statistically significant positive slope after a statistically significant negative slope. Symptom recovery was defined as symptom severity of $\leq 3$ in two contiguous measurements. The accuracy of pain severity on days 1–5 for predicting pain recovery was determined using area under the receiver operating characteristic curves. We applied Cox proportional hazards models for multivariate analyses of the potential predictors of early pain recovery. The median age was 70 years, and females accounted for 48%. The median interval from surgery to the first post-discharge clinic visit was 20 days. Trajectories of several core symptoms including pain showed a rebound from day 3 or 4. Specifically, pain severity in patients with unrecovered pain had been higher than those with recovered pain since day 4. Pain severity on day 4 showed the highest area under the curve of 0.723 for predicting pain recovery ($P = 0.001$). Multivariate analysis identified pain severity of $\leq 1$ on day 4 as an independent predictor of early pain recovery (hazard ratio, 2.86; $P = 0.0027$). Duration of symptom was the leading cause of postoperative distress. Several core symptoms after thoracoscopic lung resection showed a rebound in the trajectory. Specifically, a rebound in pain trajectory may be associated with unrecovered pain; pain severity on day 4 may predict early pain recovery. Further clarification of symptom severity trajectories is essential for patient-centered care.

researchers who meet the criteria for access to stored data.

**Funding:** TS received Osaka Cancer Society Grant-in-aid for Cancer Research 2019 (https://www. osakacancer.jp/). The funder had no role in study design, data collection and analysis, decision to publish, or preparation of the manuscript.

**Competing interests:** The authors have declared that no competing interests exist.

## Introduction

The use of video-assisted thoracoscopic surgery (VATS) has been increasing worldwide since its introduction in the 1990s [1, 2]. In Japan, the annual number of patients undergoing VATS was more than 40,000 in 2016, accounting for 67.2% and 93.1% of all surgeries for primary lung cancer and metastatic lung tumor, respectively [3]. With respect to individual cases, each patient with cancer experiences a myriad of physical, mental, emotional, and socioeconomic distresses [4]. Given the underestimation of patients' symptoms by clinicians, as reported in previous studies [5, 6], patient-reported outcomes (PROs) have gained prominence as essential tools for bridging the gap between the patients' experience and quality of life (QoL) and the healthcare providers' interpretation.

In thoracic surgery, several longitudinal studies have demonstrated a "decrease-after-a-peak" pattern in symptom severity trajectories in patients undergoing pulmonary resection. Briefly, symptom severities have been reported to peak immediately after pulmonary resection, followed by a continuous decline toward a mild level by 3 months and toward baseline level by 3–12 months [7–18]. However, details regarding symptom severity trajectories and distresses from VATS lung resection to the first post-discharge clinic visit (approximately the first 3 post-operative weeks), which is a critical time point when surgical patients face functional deterioration and start rehabilitation, remain to be fully elucidated. Therefore, this study aimed to better characterize patient symptom severity and distresses after VATS lung resection until the post-discharge follow-up visit.

## Materials and methods

This study was conducted in accordance with the principles outlined in the Helsinki Declaration, as revised in 2013. This study was approved by the Kansai Medical University Hospital Research Ethics Committee (approval number: 2018178; approval date: March 6, 2019) and registered at the University Hospital Medical Information Network Clinical Trials Registry (identification number: 000036985). Written informed consent was preoperatively obtained from all patients for their participation in this study.

### Patients

The CONSORT diagram illustrating the study population is presented in Fig 1A. A total of 130 patients who met the following inclusion criteria were prospectively enrolled: (1) patients aged ≥20 years; (2) patients who were scheduled for pulmonary resection for diagnosed or suspected pulmonary malignancy at Kansai Medical University Hospital between May 2019 and February 2020; and (3) patients who were willing to record their symptom severity using the Japanese version of the MD Anderson Symptom Inventory (MDASI), a 0–10 numeric rating scale for 13 core symptoms and 6 interferences in daily life [19, 20]. Among these patients, 3 were excluded because their operation resulted in non-pulmonary resection, 8 were excluded because they underwent thoracotomy, 18 were excluded because neither in-hospital nor post-discharge symptom records were obtained at follow-up, and 10 were excluded because no post-discharge symptom records were acquired at follow-up. Further, considering the association of postoperative complications or absence of regional analgesia with the severity and recovery of postoperative symptom [16, 21], 14 patients were excluded because they reported grade ≥II postoperative complications and 2 were excluded because they had neither epidural anesthesia nor paravertebral block. Postoperative complications were graded using the Ottawa Thoracic Morbidity and Mortality System [22], which was developed according to the Clavien-Dindo classification system [23]. Finally, this study included 75 patients undergoing

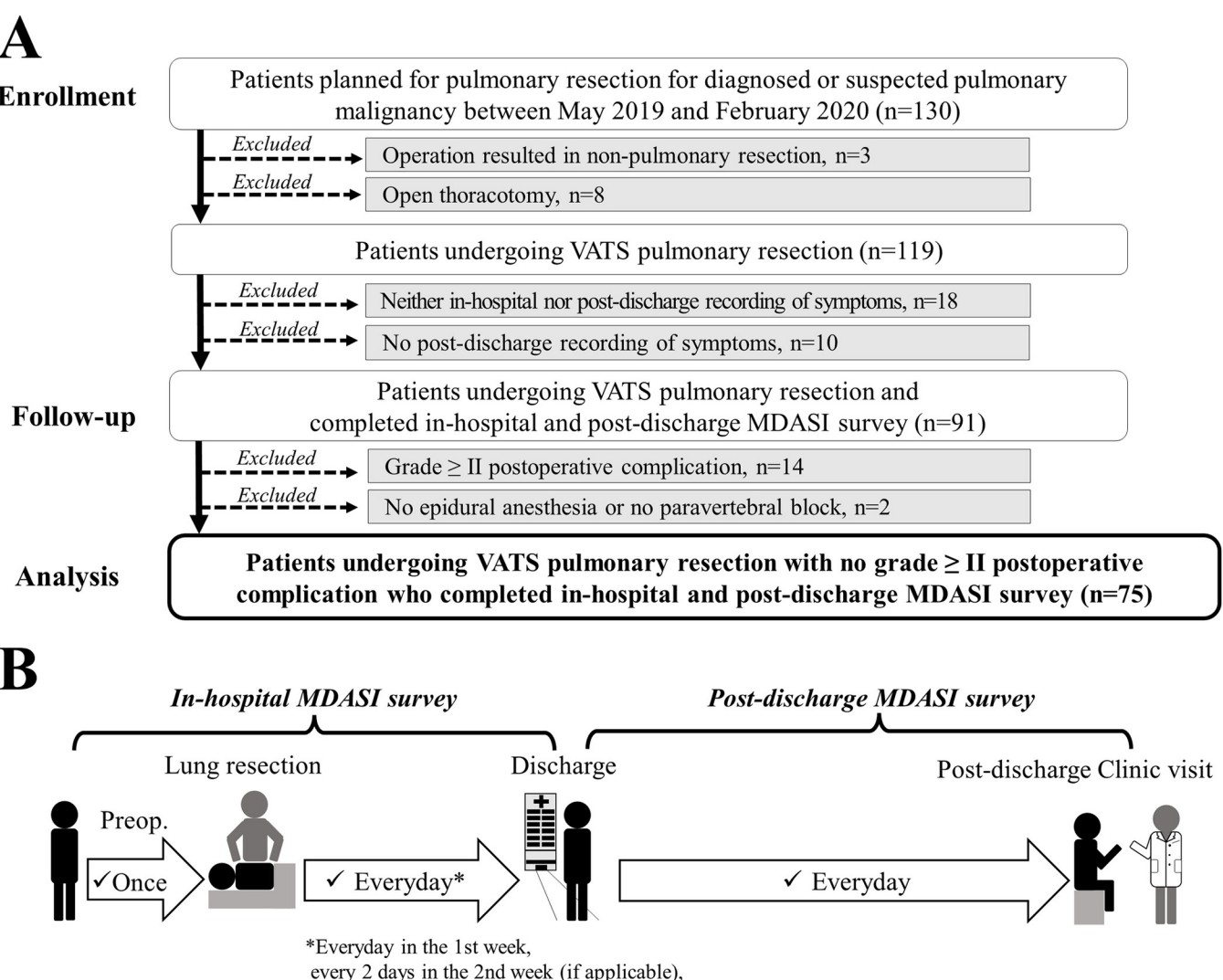

**Fig 1. CONSORT flow diagram of the study population and schedule for symptom severity recording.** A: Flow diagram depicting the selection process of participants eligible for data analysis. B: The in-hospital recording of symptom severity/interference was scheduled once preoperatively; once daily during the first postoperative week; and if applicable, every 2 days during the second postoperative week and once a week during the third postoperative week. The post-discharge recording of symptom severity/interference was scheduled daily until the first outpatient clinic visit. MDASI: MD Anderson Symptom Inventory; VATS: Video-assisted thoracoscopic surgery.

VATS pulmonary resection with no grade ≥II postoperative complications who completed in-hospital and post-discharge MDASI survey.

## Surgical procedure and anesthetic management

Surgery was performed under general analgesia with double-lumen endotracheal tube placement and epidural or paravertebral block analgesia. A two- to four-port approach was employed for VATS lobectomy or segmentectomy, whereas a three-port approach was utilized for VATS wedge resection. For patients undergoing lobectomy for primary lung cancer, lobe-specific lymph node dissection was routinely performed. For those undergoing lobectomy for metastatic lung tumor, either lobe-specific or hilar lymph node dissection was selected at the

surgeon's discretion. Hilar lymph node dissection was accompanied with segmentectomy, whereas no lymph node dissection was performed in conjunction with wedge resection. A 24- or 28-French surgical chest tube was placed upon chest closure. Epidural or paravertebral block patient-controlled analgesia was routinely achieved using the following regimen: 500 μg/ 10 mL of fentanyl plus 200 mL of 0.25% levobupivacaine and 90 mL of normal saline.

## Clinical pathway for lung resection

A clinical pathway (version date: March 26, 2018) was used for pre- and postoperative patient management. Briefly, on the day of surgery (day 0), patients were managed at our general intensive care unit, unless it was over capacity. On day 1, patients were scheduled to start physiotherapy with oral administration of 60 mg of loxoprofen thrice daily, unless they had chronic kidney disease. For patients with chronic kidney disease, 500–750 mg of acetaminophen was orally administered four times daily. During hospitalization, acetaminophen and tramadol were used as rescue analgesics, and pregabalin or mirogabalin were used for neuropathic pain at the attending surgeon's discretion. On day 2, the chest tube was removed, unless there was a sign of bleeding, air leak, or chyle leak. On day 3, the epidural or paravertebral analgesia tube was removed. In the clinical pathway, discharge was planned from day 5 to day 7. The first post-discharge follow-up clinic visit was usually scheduled at approximately 2 weeks after hospital discharge. At the time of the first post-discharge clinic visit, patients were informed about their pathological diagnosis, and their stitches on the chest tube site were removed.

## Recording of symptom severity and interference

Patients were asked to prospectively record their symptom severity and interference both during hospitalization and after hospital discharge using the printed Japanese version of the MDASI [19, 20]. The in-hospital recording of symptom severity/interference was scheduled as follows until hospital discharge: once preoperatively; daily during the first postoperative week; and if applicable, every two days during the second postoperative week, and once a week during the third postoperative week (Fig 1B). The post-discharge recording of symptom severity was scheduled daily until the first outpatient clinic visit. Regarding the post-discharge symptom report, a blank booklet of the printed Japanese version of MDASI sheets was distributed to the patients on the day of discharge, and the completed booklet was collected at the time of the first outpatient clinic visit.

## Recording of distresses and difficulties

On the day of the first post-discharge clinic visit, patients were asked to report the distresses and difficulties that they encountered during hospitalization and after hospital discharge. Moreover, they were asked to report what they would like to know more during hospitalization and after hospital discharge. For this purpose, we used questionnaires cited from those used in a previous study titled "The views of 4,054 people who faced up to cancer" conducted by the Joint Study Group on the Sociology of Cancer (S1 Text) [24].

## Statistical analysis

The average of all available scores for each symptom was calculated to estimate the degree of its impact on patients, as described previously [18]. For analysis of symptom severity trajectories, the symptom severity or symptom interference level on day 1 was compared with that on days 5, 10, 15, and 20 using paired t-tests. Further, joinpoint regression analysis was conducted to analyze the postoperative symptom severity trajectories. The number of joinpoint(s) for the

linear model was determined using the calculated Bayesian information criterion. A rebound was defined as a statistically significant positive slope after a statistically significant negative slope. For further analysis on the symptom showing a rebound, patients were subdivided into two groups based on whether their symptom recovered to a mild level by the first post-discharge outpatient clinic visit (i.e., the recovered symptom group and the unrecovered symptom group). Differences in symptom severity or symptom interference between the two groups were determined by a t-test, and joinpoint regression analysis was conducted for the respective groups.

Postoperative symptom recovery was defined according to a previous study [16]. Briefly, recovery to a mild level was defined as patient-reported symptom severity/interference of $\leq 3$ in two contiguous measurements. The time to recovery to a mild level was defined as the interval between the date of surgery and the first date showing patient-reported symptom severity/ interference of $\leq 3$. The cumulative rate of symptom recovery to mild level was calculated using the Kaplan-Meier method. As a secondary analysis, the accuracy of pain severity on days 1, 2, 3, 4, 5, 6, and 7 for predicting recovery of pain to a mild level at the first post-discharge clinic visit were determined using area under the receiver operating characteristic (ROC) curves. In this analysis, the cut-off level of pain at the earliest timepoint to predict pain recovery at the time of the first post-discharge clinic visit was also determined by the ROC curve. This was based on the concept that earlier prediction of pain recovery would give us more chances for intervention to improve the patients' outcomes. Cox proportional hazards models were used for univariate and multivariate analyses of the potential predictors of pain recovery to a mild level. Univariate analysis was used to evaluate the cut-off level of pain at the earliest timepoint determined by ROC curve; age; sex; European Cooperative Oncology Group performance status; body mass index; smoking history; %vital capacity; forced expiratory volume in one second; estimated glomerular filtration rate; Charlson comorbidity index; surgical procedure; number of ports for VATS; maximum wound length; duration of operation; amount of blood loss; chest tube size; total count of rescue analgesics use on day 0; and regular oral analgesics from day 1. Multivariate analysis included factors with $P < 0.10$ in the univariate analysis.

Data was analyzed using GraphPad Prism version 6.07 (GraphPad Software, San Diego, LA, USA), Joinpoint Regression Program version 4.9.0.0 (March 2021; Statistical Research and Applications Branch, National Cancer Institute, USA), and JMP Pro version 13.0.0 (SAS Institute, Cary, NC, USA). Statistical significance was set at $P < 0.05$.

## Results

The clinical characteristics of the study population (n = 75) are summarized in Table 1. Briefly, the median age at surgery was 70 years (range, 33–84 years), females accounted for 48.0% (36 of 75 patients), lobectomy accounted for 72.0% (54 of 75 patients), and the median duration of surgery was 102 minutes (range, 28–220 minutes). Further, the median intervals from surgery to chest tube removal, epidural or paravertebral block catheter removal, hospital discharge, and the first post-discharge clinic visit were 2, 2, 7, and 20 days, respectively. Eight (10.7%) patients had received pain medication for their chronic pain before surgery.

Clinical stages 0-II lung cancer was the most common preoperative clinical diagnosis, accounting for 80.0% (60 of 75 patients). Likewise, pathological stages 0-II lung cancer was the most common postoperative pathological diagnosis, accounting for 78.7% (59 of 75 patients). Detailed information on the patients' clinical and pathological TNM findings is shown in S1 Fig. Of note, a pathological diagnosis of lung cancer was made preoperatively in 9 patients.

Symptom scores for the 13 core symptoms and 6 interferences are summarized in S1 Table. Pain, disturbed sleep, shortness of breath, drowsiness, fatigue, and numbness and tingling

**Table 1. Clinical and pathological characteristics of the patients (N = 75).**

| Characteristics | |
|---|---|
| Age, years | 70 [33–84] |
| *Sex* | |
| Male | 39 (52.0%) |
| Female | 36 (48.0%) |
| *ECOG-PS* | |
| 0 | 68 (90.7%) |
| 1 | 6 (8.0%) |
| 3 | 1 (1.3%) |
| Body mass index | 22.7 [16.4–31.0] |
| *Smoking history* | |
| Never smoked | 26 (34.7%) |
| Ever smoked | 49 (65.3%) |
| *Pulmonary function* | |
| %vital capacity, %predicted | 99.8±15.3 |
| Forced expiratory volume in one second%, % | 72.4±10.1 |
| Estimated GFR, mL·min$^{-1}$·1.73$^{-1}$ m$^{-2}$ | 69.3±15.3 |
| *Charlson comorbidity index* | |
| 0 | 42 (56.0%) |
| 1 | 23 (30.7%) |
| 2 | 9 (12.0%) |
| 3 | 1 (1.3%) |
| *Preoperative pain medication* | 8 (10.7%) |
| For chronic pain owing to althralgia | 7/8 |
| For chronic pain following surgery for breast cancer | 1/8 |
| *Preoperative clinical diagnosis* | |
| cStage 0/ I/ II lung cancer | 60 (80.0%) |
| cStage III/ IV lung cancer | 3 (4.0%) |
| Metastatic lung tumor | 12 (16.0%) |
| *Postoperative pathological diagnosis* | |
| pStage 0/I/II lung cancer | 55 (73.3%) |
| pStage III/IV lung cancer | 4 (5.3%) |
| Metastatic lung tumor | 12 (16.0%) |
| Benign lesion | 4 (5.3%) |
| *Surgery type* | |
| Lobectomy | 54 (72.0%) |
| Segmentectomy | 1 (1.3%) |
| Wedge resection | 20 (26.7%) |
| Maximum wound length, cm | 3.5 [1.5–4.0] |
| *VATS approach* | |
| Two-port | 15 (20.0%) |
| Three-port | 19 (25.3%) |
| Four-port | 41 (54.7%) |
| Duration of operation, min | 102 [28–220] |
| Amount of blood loss, mL | 10 [0–200] |
| *Size of chest tube* | |
| 24 French | 63 (84.0%) |
| 28 French | 12 (16.0%) |
| *Intraoperative intravenous analgesics* | |

*(Continued)*

**Table 1.** (Continued)

| Characteristics | |
|---|---|
| Fentanyl, microgram | 200 [0–500] |
| Remifentanyl, mg | 0 [0–2.1] |
| *Type of regional anesthesia* | |
| EA | 58 (77.3%) |
| PVB | 17 (22.7%) |
| *Intraoperative analgesics via EA/PVB catheter* | |
| 0.25% levobupivacaine, mL | 20 [15–40] |
| 1% lidocaine, mL | 6 [0–24] |
| Fentanyl, microgram | 0 [0–100] |
| *Patient-controlled analgesics (EA/PVB)* | |
| Initial dose, mL/hr | 4 [4–6] |
| *Count of rescue analgesics use on day 0* | |
| *Total count* | 1 [0–8] |
| *Count of PCA flush* | 1 [0–8] |
| *Count of non-PCA rescue use* | 0 [0–3] |
| *Regular oral analgesics from day 1#* | |
| Loxoprofen | 66 (88.0%) |
| Daily dose of loxoprofen, mg/day | 180§ |
| Acetaminophen | 9 (12.0%) |
| Daily dose of acetaminophen, mg/day | 3,000 [1,500–4,000] |
| None | 1 (1.3%) |
| *Rescue non-PCA analgesics use from day 1 to hospital discharge#* | |
| Acetaminophen (intravenous), 1000 mg | 12 (16.0%) |
| Total instances of rescue acetaminophen use | 2 [1–7] |
| Tramadol (oral), 25 mg | 20 (26.7%) |
| Total instances of rescue tramadol use | 1 [1–11] |
| None | 47 (62.7%) |
| *Time to events, days after surgery* | |
| Chest tube removal | 2 [1–5] |
| EA/PVB catheter removal | 2 [1–5] |
| Hospital discharge | 7 [3–16] |
| The first post-discharge clinic visit | 20 [8–34] |

EA, epidural anesthesia; ECOG-PS, European Cooperative Oncology Group performance status; GFR, glomerular filtration rate; PVB, paravertebral block; VATS, video-assisted thoracoscopic surgery.

#Multiple choices were allowed.

§All 66 patients received 180 mg/day of loxoprofen.

were the six most severe symptoms. The interference with the highest symptom score was general activity.

Symptom severity trajectories for the six most severe symptoms and six interferences and their models using the joinpoint analysis are summarized in Fig 2 and S2 Table. The symptom severity of pain, disturbed sleep, drowsiness, and fatigue was significantly higher on day 1 than on days 5, 10, 15, and 20; whereas the symptom severity of shortness of breath and that of numbness and tingling on day 1 showed no significant difference compared to that on days 5, 10, 15, and 20 (Fig 2A). Further, joinpoint analysis revealed that pain, fatigue, and numbness and tingling showed a rebound in the trajectories from day 3 or 4 (Fig 2B). The interference

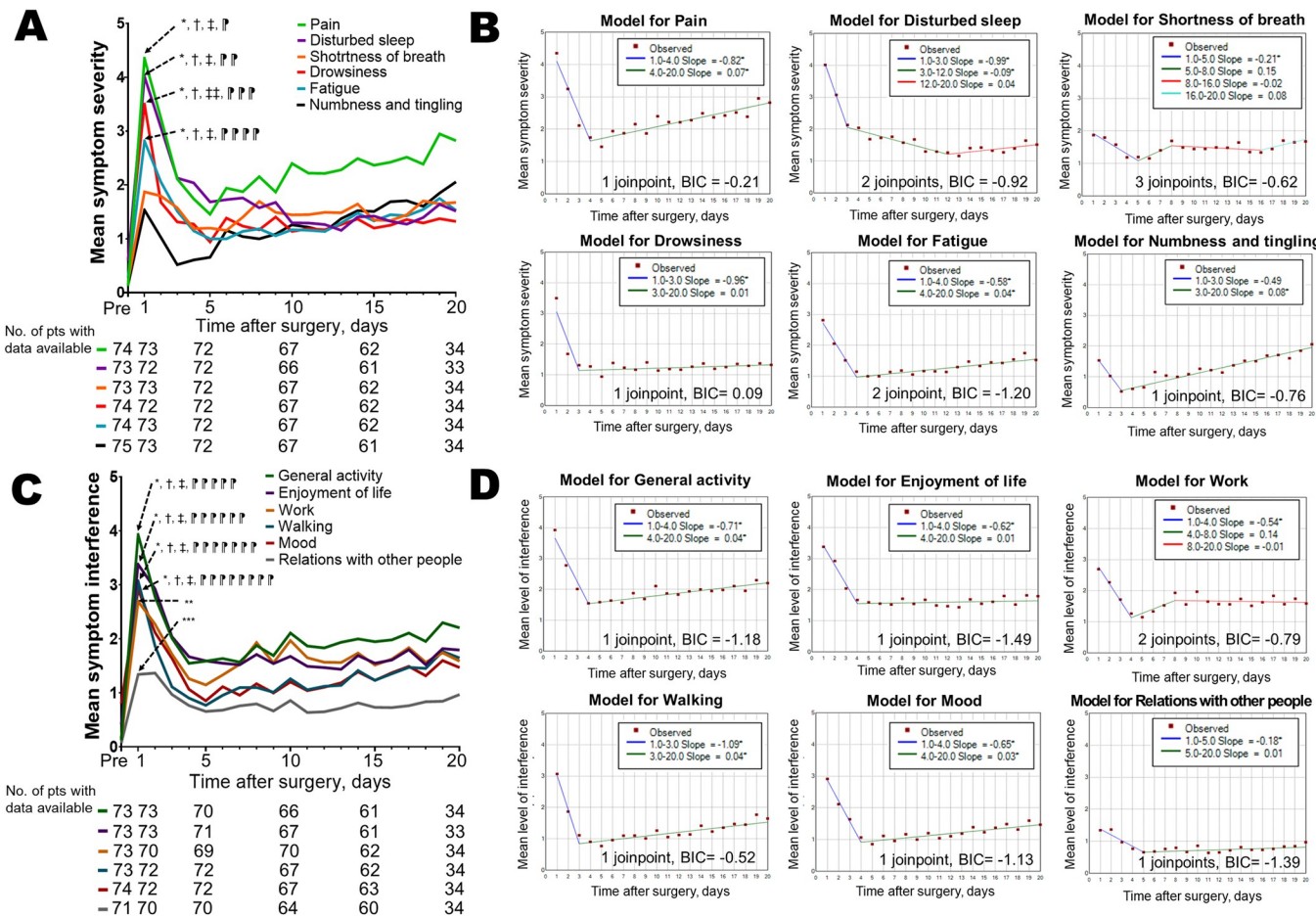

**Fig 2. Symptom severity/interference trajectories and the models using the joinpoint regression analysis.** A: The symptom severity of pain, disturbed sleep, drowsiness, and fatigue was significantly higher on day 1 than on day 5, day 10, day 15 and day 20 (pain, $P < 0.0001$, $P < 0.0001$, $P < 0.0001$, and $P = 0.017$; disturbed sleep, $P < 0.0001$ in all comparisons; fatigue, $P < 0.0001$, $P < 0.0001$, $P = 0.0002$, and P = 0.047; drowsiness, $P < 0.0001$, $P < 0.0001$, $P < 0.0001$, and $P = 0.0004$). However, the symptom severity of shortness of breath and that of numbness and tingling on day 1 showed no significant difference compared to that on day 5, day 10, day 15 and day 20 (shortness of breath, $P = 0.18$, $P = 0.74$, $P = 0.41$, and $P > 0.99$; numbness and tingling, $P = 0.19$, $P = 0.82$, $P = 0.94$, and $P = 0.20$). B: Joinpoint analysis revealed that pain, fatigue, and numbness and tingling showed a rebound in the trajectories. C: The interference level of general activity, enjoyment of life, walking and mood was higher on day 1 than on day 5, day 10, day 15, and day 20 (general activity, $P < 0.0001$, $P < 0.0001$, $P < 0.0001$, and $P = 0.0003$; enjoyment of life, $P = 0.0001$, $P < 0.0001$, $P < 0.0001$, and $P = 0.011$; walking, $P < 0.0001$, $P < 0.0001$, $P < 0.0001$, and $P = 0.0016$; mood, $P < 0.0001$, $P < 0.0001$, $P = 0.0001$, and $P = 0.016$. The symptom interference of work and relationship with other people on day 1 was higher than that on day 5, but showed no significant difference compared to that on day 10, day 15 and day 20 (work, $P = 0.0072$, $P = 0.20$, $P = 0.056$, and $P = 0.33$; relationship with other people, $P = 0.012$, $P = 0.11$, $P = 0.053$, and $P = 0.48$). D: Symptom severity trajectories of general activity, walking, and mood showed a rebound pattern.

level of general activity, enjoyment of life, walking and mood was higher on day 1 than on days 5, 10, 15, and 20; whereas the symptom interference of work and relationship with other people was higher on day 1 than only on day 5, showing no significant difference from days 10, 15, and 20 (Fig 2C). Further, trajectories of interference in general activity, walking, and mood showed a rebound pattern from day 3 or 4 (Fig 2D).

For symptoms/interferences showing a rebound, symptom severity trajectories and the models using the joinpoint analysis according to the symptom recovery status at the first post-discharge outpatient clinic (i.e., recovered symptom or unrecovered symptom) are shown in Fig 3. Patients with unrecovered pain had been experiencing more severe pain than those with recovered pain since day 4. Both of patients with recovered pain and those with unrecovered

pain showed a rebound in the trajectories (Fig 3A). Similarly, patients with unrecovered fatigue tended to have been experiencing more severe fatigue than those with recovered fatigue since day 4. Both of patients with recovered fatigue and those with unrecovered fatigue showed a rebound in the trajectories (Fig 3B). Numbness and tingling were more severe in patients with recovered numbness and tingling than those with unrecovered numbness and tingling on day 1. Since day 7, patients with unrecovered numbness and tingling had been experiencing more severe numbness and tingling than their counterparts. Only patients with unrecovered numbness and tingling showed a rebound in the trajectory (Fig 3C). Patients with unrecovered general activity had been experiencing more interferece with general activity than those with recovered general activity since day 4. Both of patients with recovered general activity and those with unrecovered general activity showed a rebound in the trajectories (Fig 3D). Walking was more interfered in patients with recovered walking than those with unrecovered walking on day 1. Since day 3, the interference level of walking had been higher in patients with unrecovered walking than those with recovered walking. Patients with recovered walking showed a rebound, whereas those with unrecovered walking showed a consistent positive slope over time in the trajectory (Fig 3E). Mood had been more interfered in patients with unrecovered mood than those with recovered mood since day 3. Only patients with unrecovered mood showed a rebound in the trajectory (Fig 3F).

Recovery of the core symptoms and interferences to mild level by the first post-discharge outpatient clinic are summarized in Fig 4. The 20-day cumulative recovery to a mild level for pain, disturbed sleep, drowsiness, fatigue, shortness of breath, and numbness and tingling were 55.8%, 82.0%, 72.6%, 67.2%, 63.7% and 33.9%, respectively (Fig 4A). Additionally, the 20-day cumulative recovery to a mild level for general activity, enjoyment of life, work, walking, mood, and relationship with other people were 72.6%, 64.1%, 50.1%, 71.6%, 63.4%, and 74.0%, respectively (Fig 4B). The distribution of symptom recovery status at the first post-discharge outpatient clinic visit showed patients with unrecovered pain were associated with increased number of unrecovered symptoms or interferences than those with recovered pain (mean number of unrecovered symptoms or interferences, 5.8 vs 1.9, $P < 0.0001$, Fig 4C).

The ROC curves of pain severity level on days 1, 2, 3, 4, 5, 6, and 7 for predicting recovery of pain to mild level at the time of the first post-discharge clinic visit are shown in Fig 5. The area under the curve (AUC) of pain severity level on day 1, 2, 3, 4, 5, 6, and 7 was 0.502, 0.549, 0.626, 0.723, 0.755, 0.842, and 0.823 ($P = 0.98$, $P = 0.48$, $P = 0.069$, $P = 0.001$, $P < 0.001$, $P < 0.001$, and $P < 0.001$), respectively. Thus, day 4 was identified as the earliest timepoint to predict pain recovery at the time of the first post-discharge clinic visit. Further, among pain severity on day 4, scale 1 was determined as the best cut-off level to predict pain recovery to mild level by the first post-discharge clinic visit.

Univariate and multivariate Cox proportional hazards models for recovery of pain to mild level are summarized in Table 2. Univariate analysis showed pain level of 0–1 on day 4, smoking history (ever-smoked or never-smoked), surgical procedure (lobectomy or wedge resection), maximum wound length, and total count of rescue analgesics use on day 0 as potential predictors for recovery of pain. Of note, the number of ports and the type of regional anesthesia (EA or PVB) showed no statistically significant association with recovery of pain to a mild level. Finally, multivariate analysis identified pain level of 0–1 on day 4 as an independent predictor for recovery of pain with hazard ratio of 2.86 (95% confidence interval, 1.43–6.17, $P < 0.0027$).

The data on patient-reported distresses and information needs during hospitalization and after discharge are summarized in Fig 6. Overall, 67 (89.3%) patients responded to the questionnaire on distress during the hospitalization: severity of symptoms, burden on the partner, and difficulties in management of symptoms were the top three distresses, accounting for

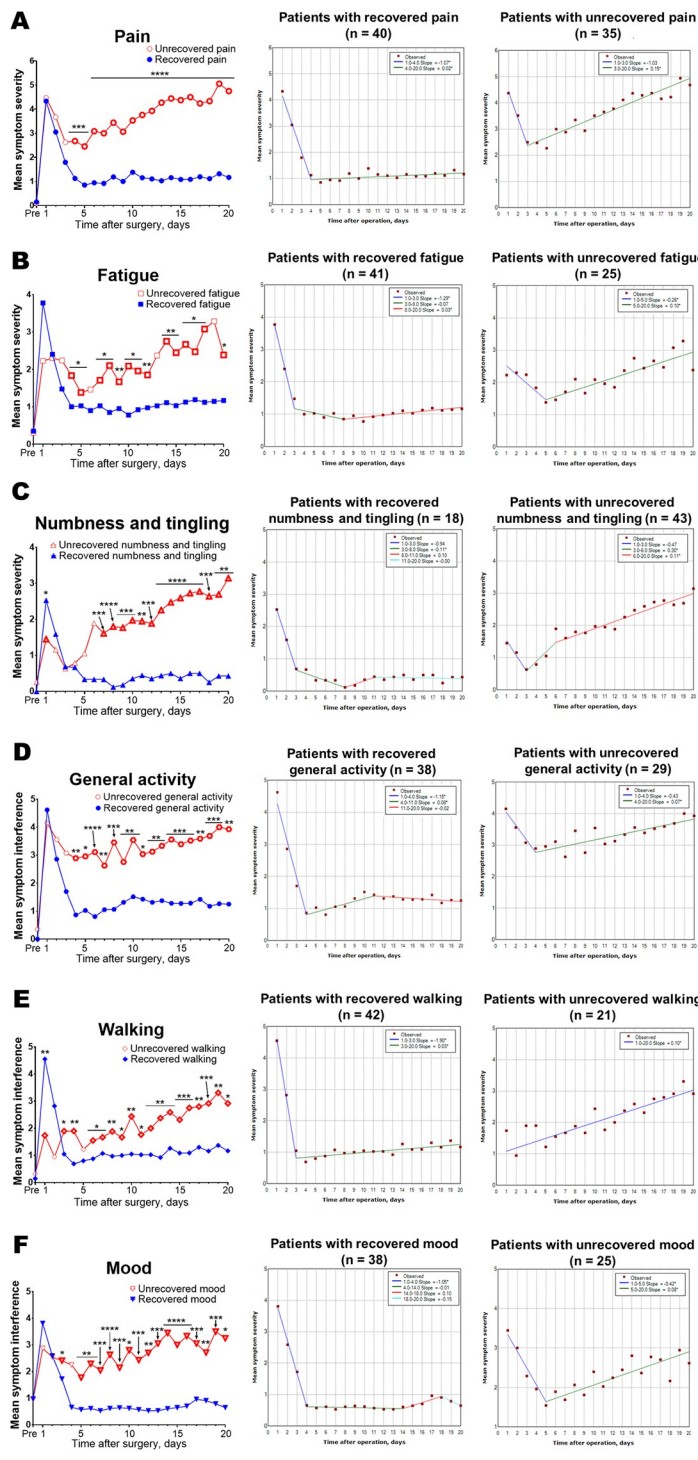

**Fig 3. Symptom severity trajectories and the models using the joinpoint regression analysis for patients with recovered or unrecovered symptom at the first post-discharge clinic visit.** A: Patients with unrecovered pain had been experiencing more severe pain than those with recovered pain since day 4. Both, patients with recovered pain and those with unrecovered pain, showed a rebound in the severity trajectory. B: Patients with unrecovered fatigue tended to experience more severe fatigue than those with recovered fatigue following day 4. Both patients with recovered fatigue and those with unrecovered fatigue showed a rebound in the severity trajectory. C: Initially, numbness and tingling were more severe in patients with recovered numbness and tingling than those with unrecovered numbness and tingling. Following day 7, patients with unrecovered numbness and tingling experienced greater numbness and tingling than their counterparts. Only patients with unrecovered numbness and tingling showed a rebound in the

severity trajectory. D: Patients with unrecovered general activity encountered greater interference than those with recovered general activity after day 4. Both, patients with recovered general activity and those with unrecovered general activity presented a rebound in the interference trajectory. E: Patients with recovered walking encountered greater interference than those with unrecovered walking on day 1. Following day 3, the interference rose higher among patients with unrecovered walking than those with recovered walking. Patients with recovered walking showed a rebound, whereas those with unrecovered walking showed a consistent positive slope over time in the interference trajectory. F: Unlike patients with recovered mood, those with unrecovered mood had been experienced a rebound and more severe interference in daily life since day 3. Patients with recovered mood did not present a rebound in the interference trajectory.

32.8%, 23.9%, and 20.9%, respectively (Fig 6A). Of note, 31.3% of the patients responding to the questionnaire reported no distress during the hospitalization. Twenty-one (28.0%) patients that responded to the questionnaire on the information needed during the hospitalization with information on symptom duration was the most frequently reported need (Fig 6B). Overall, 50 (66.7%) patients responded to the questionnaire regarding distress after hospital discharge: duration of symptoms, decrease in physical strength, and impact of symptoms on daily life were the three major sources of patient distress, accounting for 70.0%, 54.0%, and 50.0% of responses, respectively (Fig 6C). Only 8.0% of the patients that responded to the questionnaire reported no distress after hospital discharge. Seventeen (22.7%) patients provided questionnaire responses regarding the information needed after hospital discharge: duration of symptoms was the most common information need for patients (Fig 6D).

## Discussion

To the best of our knowledge, this study is based on the largest number of timepoints of an MDASI survey within the first 20 postoperative days after VATS lung resection. Our results indicate that symptom severity trajectories after VATS lung resection do not follow a uniform "decrease-after-a-peak" pattern, but involve at least two types of patterns: a rebound and a non-rebound pattern. Further, the duration of postoperative symptoms was found to be the main cause of patients' distress and the leading information need.

To date, there are only nine studies on PROs of patients undergoing VATS lung resection with more than 1 timepoint in the first postoperative week (Table 3) [11–18, 25]. Among these, four reports have described a "decrease-after-a-peak" pattern [13–16], whereas four reports, including our report, have indicated a rebound pattern [17, 18, 25]. Notably, the number of timepoints of PRO recording within the first postoperative week was 1–3 in the reports describing only a "decrease-after-a-peak" pattern, and 7 in the recent reports indicating a rebound pattern, respectively. Reportedly, a rebound in symptom severity trajectory was observed between week 1 and week 2 [17, 18]. More specifically, one study showed a rebound starting from day 4–5 [25], which matches our results. These findings suggest that a rebound in symptom severity trajectories is a phenomenon that could only be captured by close postoperative follow-up.

The underlying mechanisms that cause a rebound in symptom severity trajectories after VATS lung resection remain unknown. We found that the patients with unrecovered pain experienced greater rebound in symptom severity from day 4 in comparison to that noted in patients with recovered pain (Fig 3A). Notably, recent reports suggested the heterogeneity in pain intensity and pain recovery after surgery [26, 27], which might be one of the patient-dependent factors associated with a rebound. Interestingly, the patients with recovered pain also showed a slight rebound in the pain trajectory, which indicates a potential patient-independent factor that potentiates the increase in symptom severity (Fig 3A). In a recent study, patients undergoing thoracotomy pulmonary resection experience rebound pain after cessation of epidural analgesia, which seems predominant in male individuals [28]. In addition,

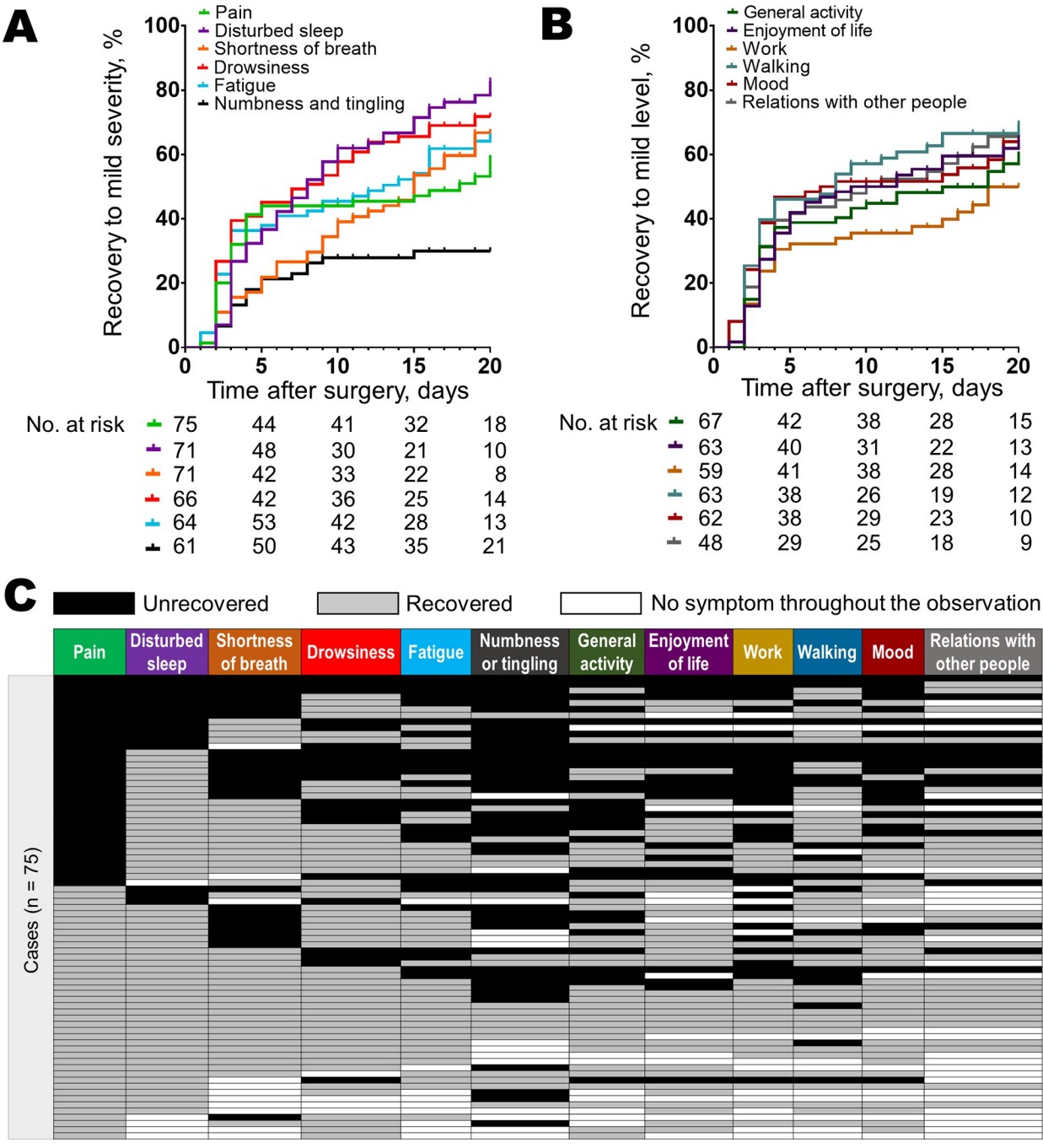

**Fig 4. Symptom recovery to mild severity and distribution of symptom recovery status at the first post-discharge outpatient clinic visit.** A: The cumulative rates of symptom recovery to a mild level of pain, interfered general activity, disturbed sleep, interfered enjoyment of life, and interfered work were 57.6%, 53.4%, 82.0%, 64.1%, and 50.2%, respectively. B: The cumulative rates of symptom recovery to a baseline level of pain, interfered general activity, disturbed sleep, interfered enjoyment of life, and interfered work were 16.4%, 20.4%, 39.7%, 38.3%, and 28.2%, respectively. C: The distribution of symptom recovery status at the first post-discharge follow-up visit revealed a higher mean number of unrecovered symptoms/ interferences in patients with unrecovered pain, than those with recovered pain (5.8 vs 1.9, respectively, $P < 0.0001$).

there are reports indicating rebound pain after peripheral nerve block wears off [29, 30]. Thus, rebound pain after cessation of regional anesthesia might be a possible patient-independent factor explaining a rebound in the pain trajectory.

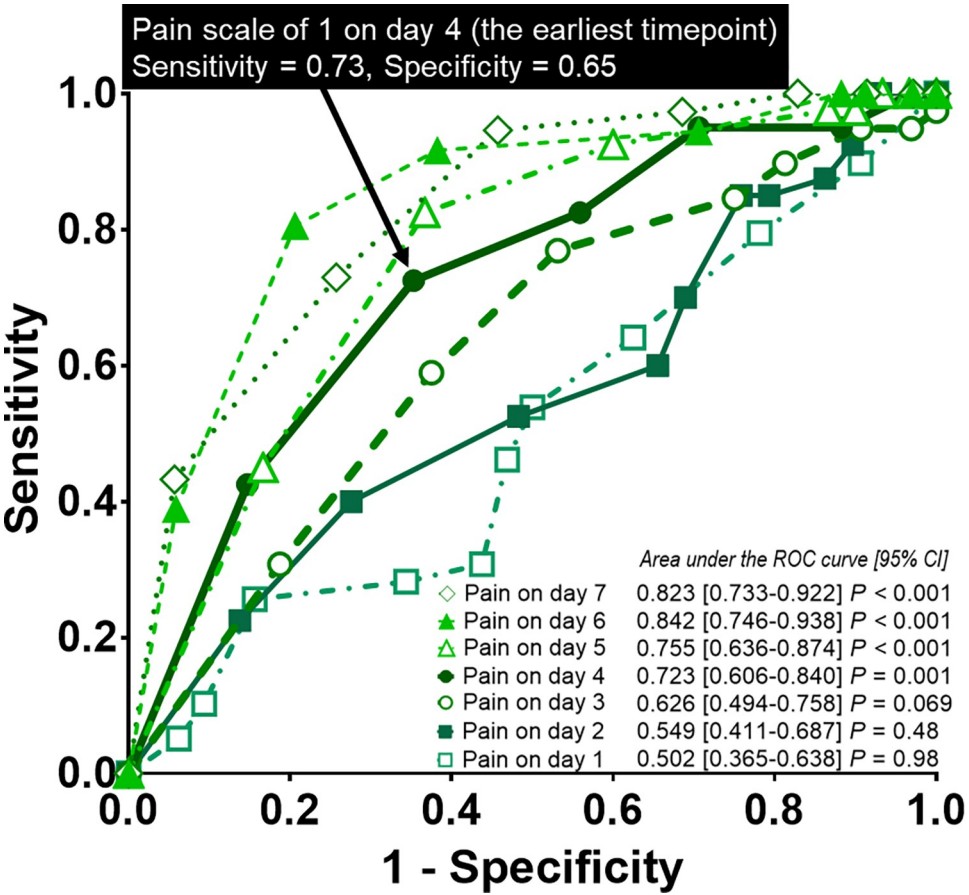

**Fig 5. The receiver operating characteristic (ROC) curves of pain severity level on days 1, 2, 3 and 4 for predicting recovery of pain to mild level at the time of the first post-discharge clinic visit.** The area under the ROC curve of pain severity level on day 1, 2, 3, and 4 were 0.502($P = 0.98$), 0.549 ($P = 0.48$), 0.626 ($P = 0.069$), and 0.723 ($P = 0.001$), respectively. A pain scale value of 1 on day 4 was determined as the best cut-off level to predict symptom recovery to a mild level by the first post-discharge clinic visit.

Unrecovered pain, or persistent pain is undoubtedly a major cause of postsurgical distresses and is referred to as chronic postsurgical pain if it lasts for 3 months or longer [31, 32]. Chronic postsurgical pain is reported to affect 10–30% of patients undergoing VATS lobectomy [12, 13, 33]. The underlying mechanism of pain chronification remains to be fully elucidated; molecularly, imbalance or "disequilibrium" between pronociceptive and antinociceptive systems in the periphery, spine, and supraspinal structures such as the periaqueductal gray has been suggested [34, 35]. Interestingly, chronic pain and memory share the anatomical sites of synaptic plasticity such as the limbic-cortical pathways [36]. Further, neuroimaging studies have suggested that the chronification of pain may result from persistence of pain memory via spatiotemporal reorganization of the neocortex [36]. Importantly, our secondary analysis identified pain severity (scale of 0–1) on day 4 as a potentially modifiable factor associated with earlier recovery of pain (Table 2). This may indicate the possible association of delayed recovery of acute postsurgical pain with persistence of pain memory. It may also indicate the benefit of early recognition of persistent pain (scale $\geq$ 2) and aggressive pain management targeting early pain relief. Considering the possible association of pain with other postoperative symptoms and interferences, improved management of pain and other symptoms in the early recovery phase would be pertinent to achieve better postoperative QoL.

**Table 2. Univariate and multivariate Cox proportional hazards models for recovery of pain to mild level (N = 75).**

| Characteristics | Univariate Cox Proportional Hazards Model | | | | Multivariate Cox Proportional Hazards Model | | | |
|---|---|---|---|---|---|---|---|---|
| | Hazard Ratio | 95% CI | P-Value | Overall P-Value | Hazard Ratio | 95% CI | P-Value | Overall P-Value |
| Pain level on day 4: 0–1 vs. 2–10 | **3.05** | **1.55–6.44** | **0.0010** | | **2.86** | **1.43–6.17** | **0.0027** | |
| Age | 0.99 | 0.96–1.02 | 0.40 | | | | | |
| Sex: female vs. male | 0.94 | 0.50–1.76 | 0.83 | | | | | |
| ECOG-PS: 1–3 vs. 0 | 1.50 | 0.44–3.81 | 0.47 | | | | | |
| Body mass index | 1.02 | 0.92–1.12 | 0.75 | | | | | |
| Smoking history: ever-smoked vs. never-smoked | **1.92** | **0.96–4.15** | **0.064** | | 1.66 | 0.82–3.65 | 0.16 | |
| Vital capacity, % | 0.99 | 0.97–1.01 | 0.55 | | | | | |
| Forced expiratory volume in one second, % | 1.01 | 0.98–1.05 | 0.49 | | | | | |
| eGFR, mL·min$^{-1}$·1.73$^{-1}$ m$^{-2}$ | 1.00 | 0.98–1.02 | 0.78 | | | | | |
| Charlson comorbidity index: 2–3 vs. 0–1 | 1.70 | 0.69–3.66 | 0.23 | | | | | |
| Surgery type: lobectomy vs. wedge resection | **0.54** | **0.28–1.10** | **0.087** | | 0.46 | 0.16–1.35 | 0.16 | |
| Maximum wound length, cm | **0.40** | **0.23–0.68** | **0.0011** | | 1.11 | 0.57–2.18 | 0.76 | |
| Number of ports for VATS | | | | 0.45 | | | | |
| Four | 1 | | | | | | | |
| Three | 1.49 | 0.66–3.16 | 0.32 | | | | | |
| Two | 1.54 | 0.66–3.35 | 0.30 | | | | | |
| Duration of operation, min | 0.99 | 0.99–1.00 | 0.15 | | | | | |
| Amount of blood loss, mL | 1.00 | 0.98–1.00 | 0.33 | | | | | |
| Chest tube size: 28 french vs. 24 french | 1.20 | 0.49–2.57 | 0.67 | | | | | |
| *Postoperative pathological diagnosis* | | | | 0.76 | | | | |
| Lung cancer | 1 | | | | | | | |
| Metastatic lung tumor | 0.91 | 0.31–2.14 | 0.84 | | | | | |
| Benign lesion | 1.55 | 0.37–4.39 | 0.50 | | | | | |
| Type of regional anesthesia | | | | | | | | |
| PVB vs. EA | 1.24 | 0.57–2.54 | 0.57 | | | | | |
| Total count of rescue analgesics used on day 0 | **0.80** | **0.64–0.95** | **0.012** | | 0.84 | 0.68–1.05 | 0.10 | |
| *Regular oral analgesics from day 1* | | | | 0.49 | | | | |
| Loxoprofen | 1 | | | | | | | |
| Acetaminophen | 1.29 | 0.44–3.03 | 0.60 | | | | | |
| None | 5.58 x 10$^{-9}$ | 0–3.54 | 0.29 | | | | | |

ECOG-PS, European Cooperative Oncology Group performance status; EA, epidural anesthesia; eGFR, estimated glomerular filtration rate; PVB, paravertebral block; VATS, video-assisted thoracoscopic surgery.

The driving force of patient-centered care is a patient's specific health needs and desired health outcomes [37]. Undoubtedly, patient-centered care and co-creation of care are associated with patients' satisfaction and well-being [38]. Healthcare providers may need to capture and take advantage of patients' voiceless voices to understand the patients' unmet needs. In our study, the duration of symptoms was a major source of patients' distress and unmet information needs (Fig 6). Further efforts in the prospective accumulation and rigorous analysis of PRO measures are crucial to further update our understanding of patients' postoperative recovery that should be shared with patients and ultimately to lead co-creation of patient-centered postoperative care. Digitalization and artificial intelligence technologies would aid in these processes [39].

This study has some limitations. First, our results were based on single-institution data and a small sample size. We acknowledge the possibility that patient characteristics and clinical practices influenced our results. The 20-day (3-week) cumulative rate of recovery of pain to

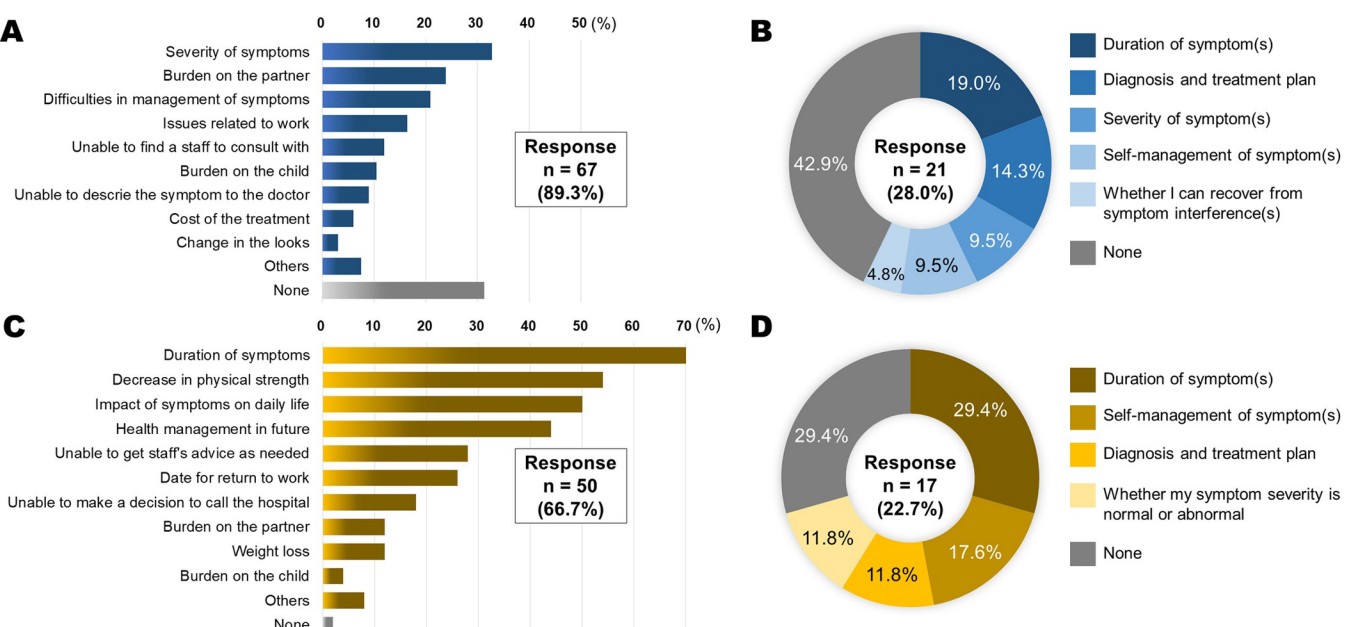

**Fig 6. Reported patients' distresses and information needs during hospitalization and after hospital discharge.** A: Sixty-seven (89.3%) patients responded to the questionnaire for distress during the hospitalization; severity of symptoms, burden on the partner, and difficulties in management of symptoms were the three major distresses, accounting for 32.8%, 23.9%, and 20.9%. Of note, 31.3% of the questionnaire responses reported no distress during hospitalization. B: Twenty-one (28.0%) patients responded on concerning information needed during the hospitalization: duration of the symptoms was the leading concern accounting for 19.0% of participants. C: For distress after hospital discharge, 50 (66.7%) patients responded the questionnaire: duration of symptoms, decrease in physical strength, and impact of symptoms on daily life were the top three distresses, accounting for 70.0%, 54.0%, and 50.0%, respectively. No distress after hospital discharge was reported by only 8.0% of the patients responded the questionnaire. D: Seventeen (22.7%) patients responded the questionnaire on the information need after hospital discharge: duration of symptom was the leading concern accounting for 29.4%.

mild severity was relatively lower than that reported in the previous reports [16, 17], for which the possible explanations include the difference in patient characteristics and in the pain management protocol. However, the 20-day (3-week) cumulative rate of recovery of general activity to baseline was similar with that noted in the previous report [14]. Thus, our results may not completely contradict the findings of previous reports. Further, our follow-up rate in the presented study was 76.4% (91 of 119 patients), which was comparable to the rate (77.9%, 60 of 77 patients) in a previous report [16]. To validate our findings, further studies should include a larger study population from multiple institutions. Second, our study population had heterogenous clinical and pathological characteristics. Patients with metastatic lung cancer/benign pathology and those undergoing sublobar resection were included in our study, which could result in heterogeneity in the extent of pulmonary resection and lymph node dissection. This could impact postoperative symptoms. Additionally, 9 of 60 patients with a clinical diagnosis of lung cancer had a pathological confirmation preoperatively. Postoperative symptoms, such as disturbed sleep, distress, mood, and pain, could be affected by not only the clinical diagnosis itself, but also the type of clinical diagnosis (i.e., pathologically-confirmed or not). Thus, adjustment for the clinical diagnosis and its type should be considered in future studies.

Third, we employed the MDASI to assess the QoL of patients. The MDASI is a simple and widely accepted PRO and QoL instrument; nonetheless, it has disadvantages (for example, it lacks information on the location of pain). It remains controversial which of PRO instruments is the best for patients undergoing VATS pulmonary resection, but the MDASI seemed at least feasible as our patients evaluated the MDASI as easy to input (n = 50, median scale 8 [11-scale with 0 as very difficult and 10 as very easy], S2 Fig). Forth, the clinical pathway employed during the study period did not correspond to up-to-date Enhanced Recovery After Surgery

**Table 3. Summary of longitudinal studies reporting PROs of patients undergoing VATS for lung resection with ≥ 1 timepoint within the first postoperative week [12–18, 25].**

| Author, Publication year | Instrument used in the study | No. of pts (VATS) | Procedure type and analgesia | Grade ≥ 2 postop. morbidity | Timeframe (Timepoints in the 1st postop. month) | Select findings in symptom severity trajectory (in *italics*) and remarks in symptom recovery |
|---|---|---|---|---|---|---|
| Bendixen, et al. 2016 [12] | EORTC QLQ-C30, NRS | 201 (102) | L + EA | Included | Preop. to 1 year (Days 1–2; Week 2, 4) | *No specific description on symptom severity trajectory*; Proportion of pts with moderate-to-severe pain decreased over time; QoL recovered to baseline by ~8 weeks in VATS group |
| Rizk, et al. 2014 [13] | SF36, BPI | 206 (132) | L, S + EA | Included | Preop. to 1 year (Days 2–4; Week 2) | *Decrease-after-a-peak* pattern (pain) from day 2 to day 4; Proportion of pts with adjusted pain ≥ 4 decreased over time |
| Shi, et al. 2016 [14] | MDASI, SF12 | 72 (32) | L + (N/A) | Included | Preop. to 3 months (Day 3, 7; Month 1) | *Decrease-after-a-peak* pattern (e.g., general activity); General activity recovered to baseline in ~18% of pts at 3 weeks |
| Xu, et al. 2020 [15] | EORTC QLQ-C30, QLQ-LC13 | 120 (120) | L + (N/A) | Included | Preop. to 8 weeks (Week 1, 2, 4) | *Decrease-after-a-peak* pattern (pain, fatigue); Physical function recovered to 80–90% of baseline by 8 weeks |
| Fagundes, et al. 2015 [16] | MDASI | 60 (29) | L + (N/A) | Included | Preop. to 3 months (Day 3, 5, 7; Month 1) | *Decrease-after-a-peak* (e.g., pain) and *Rebound* (possible) (drowsiness and SOB from day 5 to day 7); Pain recovered to mild level in ~60% (~75% in VATS) of pts at 3 weeks |
| Wei, et al., 2021 [17] | MDASI-LC | 117 (63) | L, S, W + (N/A) | Included | Preop. to 4 weeks (Days 1–7; Week 2, 3, 4) | *Rebound* (possible) (e.g., pain and work) from week 1 to week 2; Pain recovered to mild level in ~88% of pts at 3 weeks |
| Dai, 2021 [18] | MDASI-LC | 110 (110) | L, S + ICNB | Included | Preop. to 4 weeks (Days 1–7; Week 2, 3, 4) | *Rebound* (possible) (pain, disturbed sleep, SOB, fatigue, general activity, work, mood) from week 1 to week 2 |
| Chang, et al. 2022 [25] | NRS | 635 (635) | L, S, W + EA | Excluded | Preop. to day 7 (Days 1–7) | *Rebound* (possible) (pain) from day 4; Heterogeneity in pain trajectory (mild vs. rebound pattern) |
| Present study, 2021 | MDASI | 75 (75) | L, W + EA, PVB | Excluded | Preop. to day 20[¶] (Days 1–20) | *Rebound* (e.g., pain and general activity from day 5) *and non-rebound* (e.g., disturbed sleep); Pain recovered to mild level in 56% of pts at 3 weeks[¶¶]; General activity recovered to baseline in 22% of pts at 3 weeks[¶¶] |

[¶]Median follow-up time

[¶¶]Day 20. BPI, Brief Pain Inventory; EA, epidural anesthesia; EORTC, European Organization for Research and Treatment of Cancer; L, lobectomy; MDASI, MD Anderson Symptom Inventory; MDASI-LC, MD Anderson Symptom Inventory for lung cancer; NRS, numeric rating scale; PRO, patient-reported outcome; PVB, paravertebral block; QLQ-C30, Quality of Life Questionnaire Core 30; S, segmentectomy; SF, Short Form Health Survey; SOB, shortness of breath; VATS, video-assisted thoracoscopic surgery; W, wedge resection.

(ERAS) protocols. As implementation of ERAS pathways has been shown to be associated with improved patient outcomes [40], our results should be validated as part of a clinical pathway that conforms to recent ERAS guidelines [41].

In conclusion, patients undergoing VATS lung resection may experience a rebound in symptom severity trajectories of several core symptoms such as pain and general activity. Specifically, a rebound in pain trajectory may be associated with unrecovered pain; pain severity on day 4 may predict early pain recovery. Additionally, the duration of postoperative symptoms may be the main cause of the patients' distress. Thus, further clarification of symptom severity trajectories would be essential for patient-centered care.

## Supporting information

**S1 Text. Questionnaire on distresses used in this study.**
(DOCX)

**S1 Fig. Clinical and pathological TNM in patients with a diagnosis of lung cancer.**
(DOCX)

**S2 Fig. Summary of the patients' opinions regarding usefulness and feasibility of the MDASI survey (response to the Q5 in the questionnaire used in this study).**
(DOCX)

**S1 Table. Summary of symptom scores for 13 core symptoms and 6 interferences with daily life.**
(DOCX)

**S2 Table. Bayesian information criterion for models of symptom severity trajectories calculated using the joinpoint analysis.**
(DOCX)

## Acknowledgments

The authors would like to thank all members of the SMILE-001 study investigators. The full membership list of investigators in the SMILE-001 study are as follows: Tomohito Saito; Anna Hamakawa; Hideto Takahashi; Yukari Muto; Miku Mouri; Makie Nakashima; Natsumi Maru; Takahiro Utsumi; Hiroshi Matsui; Yohei Taniguchi; Haruaki Hino; Emi Hayashi; Tomohiro Murakawa; Kyoko Uehara; Akemi Takatani; Ayako Kushioka; Mariko Nishimoto; Hazuki Yamada; Mai Takiishi; Ayaka Kitagawa; Kana Shimamoto; Mana Iida; Risa Takeyasu; Hasumi Aono; Saya Terada; Yuki Tsuji; Minako Matsunobu; Minami Tadokoro; Yuko Kajiwara; Yukiko Sakai; Kie Inoue; Yuha Yasuda; Kurumi Kawai; Ami Shimazaki; Asuka Shono; Rumi Ue; Osamu Okada; Chisato Fujino; Miki Yabuuchi; Chinatsu Takahira; Natsumi Azuma; Chinatsu Hirata; Sara Kajihara; and Anna Abe.

The authors would like to express our gratitude to Ms. Mutsumi Ishikawa and Dr. Ken Yamaguchi (Shizuoka Cancer Center, Shizuoka, Japan) for providing the questionnaire for patients' difficulties and concerns. The authors would also like to thank Editage (www.editage.com) for English language editing.

## Author Contributions

**Conceptualization:** Tomohito Saito.

**Data curation:** Tomohito Saito.

**Formal analysis:** Tomohito Saito.

**Funding acquisition:** Tomohito Saito.

**Investigation:** Tomohito Saito, Anna Hamakawa, Yukari Muto, Miku Mouri, Makie Nakashima.

**Methodology:** Tomohito Saito, Hideto Takahashi.

**Project administration:** Anna Hamakawa.

**Resources:** Tomohito Saito, Anna Hamakawa, Yukari Muto, Miku Mouri, Makie Nakashima, Natsumi Maru, Takahiro Utsumi, Hiroshi Matsui, Yohei Taniguchi, Haruaki Hino.

**Supervision:** Emi Hayashi, Tomohiro Murakawa.

**Writing – original draft:** Tomohito Saito.

**Writing – review & editing:** Tomohito Saito, Anna Hamakawa, Hideto Takahashi, Yukari Muto, Miku Mouri, Makie Nakashima, Natsumi Maru, Takahiro Utsumi, Hiroshi Matsui, Yohei Taniguchi, Haruaki Hino, Tomohiro Murakawa.

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
