## [Decision Letter · Decision Letter 0]

14 Mar 2022

PONE-D-22-03682Symptom severity trajectories and distresses in patients undergoing video-assisted thoracoscopic lung resection from surgery to the first post-discharge clinic visitPLOS ONE

Dear Dr. Saito,

Thank you for submitting your manuscript to PLOS ONE. After careful consideration, we feel that it has merit but does not fully meet PLOS ONE’s publication criteria as it currently stands. Therefore, we invite you to submit a revised version of the manuscript that addresses the points raised during the review process.

Please comment on ALL issues the reviewer raised and redo your manuscript accordingly.

We look forward to receiving your revised manuscript.

Kind regards,

Eric D. Roessner, Prof.

Academic Editor

PLOS ONE

Journal Requirements:

3. One of the noted authors is a group or consortium the SMILE-001 investigators. In addition to naming the author group, please list the individual authors and affiliations within this group in the acknowledgments section of your manuscript. Please also indicate clearly a lead author for this group along with a contact email address.

Reviewers' comments:

Reviewer's Responses to Questions

**Comments to the Author**

1. Is the manuscript technically sound, and do the data support the conclusions?

Reviewer #1: Yes

Reviewer #2: Yes

2. Has the statistical analysis been performed appropriately and rigorously? 

Reviewer #1: Yes

Reviewer #2: Yes

3. Have the authors made all data underlying the findings in their manuscript fully available?

Reviewer #1: Yes

Reviewer #2: Yes

4. Is the manuscript presented in an intelligible fashion and written in standard English?

Reviewer #1: Yes

Reviewer #2: Yes

5. Review Comments to the Author

Reviewer #1: The authors focus an important issue in evaluation of PROs in thoracic surgery; however, the results of this trial do justify only a vague conclusion. I have several concerns that are listed below:

1. The sample size is small and the study population is too heterogeneous.

2. There are no data about patients with suffered on chronic pain and had possibly pain medication before surgery.

3. There are no clear data about the histopathological tumour status of the patient (TNM) pre and postoperatively. All these factors could influence symptom severity trajectories and distress.

4. It is not clear in which amount the numbers of ports affect the main endpoint.

5. The clinical pathway is faulty and do not correspond to current ERAS pathways.

6. Pain therapy algorithm is described; however, no data are available about the amount of pain medication or additional pain medication the patients had really preserved.

7. There are too many tables, could be reduced to the essential.

Reviewer #2: This manuscript is well written with a good English. It is technically sound, and the data support the conclusions. The authors made all data underlying the findings in their manuscript fully available and the statistical analysis is performed appropriately.

However I still have some questions to the authors:

1- Do you think knowing the diagnosis before surgery (as some of your paitents did and some not), more specificly knowing that they have cancer or not, or even having a primary or secondary cancer, would effect the postoperative symptoms of the patients, such as disturbed sleep, drawsiness, distress, mood, and even pain. I think you should be commenting on these in discussion.

2- I think you should emphasize if there was any difference between the groups having a paravertebral or an epidural block.

3- I think you should also be giving information about the details of the surgical technique, as primary cancer patients probably had lymph node dissection also, apart from lobectomy, secondary cancer patients might had it too or not, and patients with a benign lesion probably had no lymph node dissection, and these details might have effected some postoperative symptoms also.

6. PLOS authors have the option to publish the peer review history of their article (what does this mean?). If published, this will include your full peer review and any attached files.

Reviewer #1: No

Reviewer #2: No

---

## [Author Response · Author response to Decision Letter 0]

19 May 2022

April 25th, 2022

Editorial Board

PLOS ONE

Dear Editors:

Please find enclosed our revised manuscript entitled “Symptom severity trajectories and distresses in patients undergoing video-assisted thoracoscopic lung resection from surgery to the first post-discharge clinic visit” (PONE-D-22-03682). We have revised the manuscript based on the reviewers’ comments. We have addressed the reviewers’ concern below in a point-by-point manner, with our responses appearing in italics. 

We are extremely grateful to the Editors and the Reviewers for their careful review and helpful suggestions. We hope that the revised manuscript meets the expectations of the Editors and Reviewers and is now suitable for publication in PLOS ONE.

Sincerely,

Tomohito Saito, MD, PhD

Department of Thoracic Surgery, Kansai Medical University Hospital

2-3-1, Shinmachi, Hirakata, Osaka 573-1191, Japan

Tel: +81-72-804-0101; Fax: +81-72-804-2865 

E-mail: saitotom@hirakata.kmu.ac.jp

AUTHORS’ RESPONSES TO COMMENTS FROM REVIEWERS

Re: PONE-D-22-03682　“Symptom severity trajectories and distresses in patients undergoing video-assisted thoracoscopic lung resection from surgery to the first post-discharge clinic visit”

Reviewer #1:

The authors focus an important issue in evaluation of PROs in thoracic surgery; however, the results of this trial do justify only a vague conclusion. I have several concerns that are listed below:

1. The sample size is small and the study population is too heterogeneous.

[Authors’ response]

In response to the Reviewer’s comment, we have revised the description of the study’s limitations in the Discussion section as follows: 

“This study has some limitations. First, our results were based on single-institution data and a small sample size. We acknowledge the possibility that patient characteristics and clinical practices influenced our results.” [Page 33, Lines 524–526 of Marked Copy]

“Second, our study population had heterogenous clinical and pathological characteristics. Patients with metastatic lung cancer/benign pathology and those undergoing sublobar resection were included in our study, which could result in heterogeneity in the extent of pulmonary resection and lymph node dissection. This could impact postoperative symptoms. Additionally, 9 of 60 patients with a clinical diagnosis of lung cancer had a pathological confirmation preoperatively. Postoperative symptoms, such as disturbed sleep, distress, mood, and pain, could be affected by not only the clinical diagnosis itself, but also the type of clinical diagnosis (i.e., pathologically-confirmed or not). Thus, adjustment for the clinical diagnosis and its type should be considered in future studies.” [Page 34, Lines 534–543 of Marked Copy]

Reviewer #1:

2. There are no data about patients with suffered on chronic pain and had possibly pain medication before surgery.

[Authors’ response]

In response to the Reviewer’s comment, we have revised Table 1. We have also added the following description to the Result section: 

“Eight (10.7%) patients had received pain medication for their chronic pain before surgery.” [Pages 11-12, Lines 263–264 of Marked Copy]

Reviewer #1:

3. There are no clear data about the histopathological tumour status of the patient (TNM) pre and postoperatively. All these factors could influence symptom severity trajectories and distress.

[Authors’ response]

In response to the Reviewer’s comment, we have revised Table 1. We have also added a supplemental figure (S3 Figure) describing the preoperative and postoperative TNM statuses. Furthermore, we have added the following statement to describe the preoperative pathological diagnosis and (postoperative) pathological stage:

“Clinical stages 0-II lung cancer was the most common preoperative clinical diagnosis, accounting for 80.0% (60 of 75 patients). Likewise, pathological stages 0-II lung cancer was the most common postoperative pathological diagnosis, accounting for 78.7% (59 of 75 patients). Detailed information on the patients’ clinical and pathological TNM findings is shown in S3 Figure. Of note, a pathological diagnosis of lung cancer was made preoperatively in 9 patients.” [Page 12, Lines 265–270 of Marked Copy]

Reviewer #1:

4. It is not clear in which amount the numbers of ports affect the main endpoint.

[Authors’ response]

As shown in the Table 4 in the original manuscript (Table 2 in the Marked Copy), no association between the number of ports and the main endpoint was detected. In response to the Reviewer’s comment, we have added the following statement to clarify this point: 

“Of note, the number of ports and the type of regional anesthesia (EA or PVB) showed no statistically significant association with recovery of pain to a mild level.” [Page 21, Lines 413–415 of Marked Copy]

Reviewer #1:

5. The clinical pathway is faulty and do not correspond to current ERAS pathways.

[Authors’ response]

 In response to the Reviewer’s comment, we have added the following statement to the Discussion section: 

“Forth, the clinical pathway employed during the study period did not correspond to up-to-date Enhanced Recovery After Surgery (ERAS) protocols. As implementation of ERAS pathways has been shown to be associated with improved patient outcomes [39], our results should be validated as part of a clinical pathway that conforms to recent ERAS guidelines [40].” [Page 34, Lines 550–554 of Marked Copy]

Reviewer #1:

6. Pain therapy algorithm is described; however, no data are available about the amount of pain medication or additional pain medication the patients had really preserved.

[Authors’ response]

In response to the Reviewer’s comment, we have revised Table 1 to clarify the amount of pain medication used and the instances of rescue pain medication administration.

Reviewer #1:

7. There are too many tables, could be reduced to the essential.

[Authors’ response]

In response to the Reviewer’s comment, we have removed two tables (Tables 2 and 3 in the original version) from the main manuscript and now provide them in the form of supporting information as S4 Table and S5 Table, respectively.

Reviewer #2: 

This manuscript is well written with a good English. It is technically sound, and the data support the conclusions. The authors made all data underlying the findings in their manuscript fully available and the statistical analysis is performed appropriately.

However I still have some questions to the authors:

1- Do you think knowing the diagnosis before surgery (as some of your patients did and some not), more specifically knowing that they have cancer or not, or even having a primary or secondary cancer, would effect the postoperative symptoms of the patients, such as disturbed sleep, drowsiness, distress, mood, and even pain. I think you should be commenting on these in discussion.

In response to the Reviewer’s comment, we have revised the Discussion section as follows:

“Second, our study population had heterogenous clinical and pathological characteristics. Patients with metastatic lung cancer/benign pathology and those undergoing sublobar resection were included in our study, which could result in heterogeneity in the extent of pulmonary resection and lymph node dissection. This could impact postoperative symptoms. Additionally, 9 of 60 patients with a clinical diagnosis of lung cancer had a pathological confirmation preoperatively. Postoperative symptoms, such as disturbed sleep, distress, mood, and pain, could be affected by not only the clinical diagnosis itself, but also the type of clinical diagnosis (i.e., pathologically-confirmed or not). Thus, adjustment for the clinical diagnosis and its type should be considered in future studies.” [Page 34, Lines 534–543 of Marked Copy]

Reviewer #2: 

2- I think you should emphasize if there was any difference between the groups having a paravertebral or an epidural block.

[Authors’ response]

In response to the Reviewer’s comment, we have revised Table 2 of the Marked Copy. We have also added the following description to the Result section:

“Of note, the number of ports and the type of regional anesthesia (EA or PVB) showed no statistically significant association with recovery of pain to a mild level.” [Page 21, Lines 413–415 of Marked Copy]

Reviewer #2: 

3- I think you should also be giving information about the details of the surgical technique, as primary cancer patients probably had lymph node dissection also, apart from lobectomy, secondary cancer patients might had it too or not, and patients with a benign lesion probably had no lymph node dissection, and these details might have effected some postoperative symptoms also.

[Authors’ response]

 In response to the Reviewer’s comment, we have added the following statement to the Method section:

“For patients undergoing lobectomy for primary lung cancer, lobe-specific lymph node dissection was routinely performed. For those undergoing lobectomy for metastatic lung tumor, either lobe-specific or hilar lymph node dissection was selected at the surgeon’s discretion. Hilar lymph node dissection was accompanied with segmentectomy, whereas no lymph node dissection was performed in conjunction with wedge resection.” [Pages 7–8, Lines 166–171 of Marked Copy]

Further, in response to the Editorial Office’s request, we have added S6 Figure to provide the data regarding feasibility of the MDASI survey. We have corrected the numbers of the patients who responded the questionnaire [Page 34, Lines 548-550 of Marked Copy].

Additionally, we recognized that the description in Table 1 required the following corrections:

 Surgery type

Lobectomy 54 (72.0%)

Segmentectomy 1 (1.3%)

 Wedge resection 20 (26.7%)

We sincerely apologize for any confusion caused.

---

## [Decision Letter · Decision Letter 1]

9 Nov 2022

PONE-D-22-03682R1Symptom severity trajectories and distresses in patients undergoing video-assisted thoracoscopic lung resection from surgery to the first post-discharge clinic visitPLOS ONE

Dear Dr. Tomohito Saito,

Thank you for submitting your manuscript to PLOS ONE. After careful consideration, we feel that it has merit but does not fully meet PLOS ONE’s publication criteria as it currently stands. Therefore, we invite you to submit a revised version of the manuscript that addresses the points raised during the review process.

  Please reply to the reviewers comments

We look forward to receiving your revised manuscript.

Kind regards,

Silvia Fiorelli

Academic Editor

PLOS ONE

Journal Requirements:

Reviewers' comments:

Reviewer's Responses to Questions

**Comments to the Author**

1. If the authors have adequately addressed your comments raised in a previous round of review and you feel that this manuscript is now acceptable for publication, you may indicate that here to bypass the “Comments to the Author” section, enter your conflict of interest statement in the “Confidential to Editor” section, and submit your "Accept" recommendation.

Reviewer #1: All comments have been addressed

Reviewer #3: (No Response)

2. Is the manuscript technically sound, and do the data support the conclusions?

Reviewer #1: Yes

Reviewer #3: Partly

3. Has the statistical analysis been performed appropriately and rigorously? 

Reviewer #1: Yes

Reviewer #3: Yes

4. Have the authors made all data underlying the findings in their manuscript fully available?

Reviewer #1: Yes

Reviewer #3: Yes

5. Is the manuscript presented in an intelligible fashion and written in standard English?

Reviewer #1: Yes

Reviewer #3: Yes

6. Review Comments to the Author

Reviewer #1: I have no further comments, all comments have been addressed. I accept the revised manuscript for publication.

Reviewer #3: This is an interesting article that identifies characteristics regarding postoperative quality of life in patients who have undergone VATS lung resection.

As I have some questions or comments, please answer these issues.

Page 19, Line 370; The 20 day cumulative recovery to a mild level for pain … were 55,8%, …

I think the recovery rate of postoperative pain 55.8% is too low, what do you think about it?

Page 20, Line 385; Further, among pain severity on day 4, …

You collected data on pain from postoperative days 1 to day 4, and stated that the value on day 4 was optimal for determining the cutoff value, but if you had collected data on day 5 as well, wouldn't it be even more predictive? Do you think it is right?

Page 21, Line 401; Fig. 5;

This curve suggests that day 3 is a better indicator than days 1 and 2, and day 4 is a better indicator than day 3. Does this also suggest that day 5 (or day 6...) may be a better predictor than day 4?

Page 32, Line 507;

How do you think the absolute value of pain intensity affected this result? Did patients with stronger pain in the early postoperative period rebound more during the chronic period?

Does this association between pain on day 4 and chronic pain relate to so-called "pain memory"?

7. PLOS authors have the option to publish the peer review history of their article (what does this mean?). If published, this will include your full peer review and any attached files.

Reviewer #1: No

Reviewer #3: **Yes: **Yoshifumi Sano

---

## [Author Response · Author response to Decision Letter 1]

18 Dec 2022

AUTHORS’ RESPONSES TO COMMENTS FROM REVIEWERS

Re: PONE-D-22-03682R1　“Symptom severity trajectories and distresses in patients undergoing video-assisted thoracoscopic lung resection from surgery to the first post-discharge clinic visit”

Reviewer #1:

I have no further comments, all comments have been addressed. I accept the revised manuscript for publication.

[Authors’ response]

We appreciate the reviewer’s careful review and helpful suggestions that have benefitted our manuscript considerably. 

Reviewer #2: 

This is an interesting article that identifies characteristics regarding postoperative quality of life in patients who have undergone VATS lung resection.

As I have some questions or comments, please answer these issues.

Page 19, Line 370; The 20 day cumulative recovery to a mild level for pain … were 55,8%, …

I think the recovery rate of postoperative pain 55.8% is too low, what do you think about it?

[Authors’ response]

 We agree with the reviewer that the 20-day (3-week) cumulative rate of recovery of pain to mild severity in our study was relatively lower than that noted in previous studies as shown in Table 3. Possible explanations for this difference might include the difference in patient characteristics and in the pain management protocol. 

 In response to the reviewer’s comment, we have included the above information in the Discussion section as follows:

“The 20-day (3-week) cumulative rate of recovery of pain to mild severity in our study was relatively lower than that reported in previous studies [16,17], for which the possible explanations include the difference in patient characteristics and in the pain management protocol.” [Pages 34–35, Lines 538–541 of the Marked Copy]

Page 20, Line 385; Further, among pain severity on day 4, …

You collected data on pain from postoperative days 1 to day 4, and stated that the value on day 4 was optimal for determining the cutoff value, but if you had collected data on day 5 as well, wouldn't it be even more predictive? Do you think it is right?

[Authors’ response]

In terms of predictive power, the pain level on days 5, 6, and 7 seemed better than that on day 4 (AUC, 0.755, 0.842, and 0.823 vs. 0.723, respectively). However, we selected day 4 for further analysis as we identified day 4 as the earliest timepoint to predict pain recovery at the time of the first post-discharge clinic visit (Figure 5). This was based on the concept that earlier prediction of pain recovery would give us more chances for intervention to improve the patients’ outcomes. 

In response to the reviewer’s comment, we have revised the Methods section to clarify the reason why we focused on the pain level at the earliest timepoint to predict pain recovery as follows: 

“As a secondary analysis, the accuracy of pain severity on days 1, 2, 3, 4, 5, 6, and 7 for predicting recovery of pain to a mild level at the first post-discharge clinic visit was determined using area under the receiver operating characteristic (ROC) curves. In this analysis, the cut-off level of pain at the earliest timepoint to predict pain recovery at the time of the first post-discharge clinic visit was also determined by the ROC curve. This was based on the concept that earlier prediction of pain recovery would give us more chances for intervention to improve the patients’ outcomes.” [Pages 10-11, Lines 236-244 of the Marked Copy]

“Univariate analysis was used to evaluate the cut-off level of pain at the earliest timepoint determined by the ROC curve” [Page 11, Lines 245-247 of the Marked Copy]

Furthermore, we have also revised Figure 5 and the Results section as follows:

“The ROC curves of pain severity level on days 1, 2, 3, 4, 5, 6, and 7 for predicting recovery of pain to mild level at the time of the first post-discharge clinic visit are shown in Fig 5. The area under the curve (AUC) of pain severity level on days 1, 2, 3, 4, 5, 6, and 7 were 0.502, 0.549, 0.626, 0.723, 0.755, 0.842, and 0.823 (P = 0.98, P = 0.48, P = 0.069, P = 0.001, P < 0.001, P < 0.001, and P < 0.001), respectively. Thus, day 4 was identified as the earliest timepoint to predict pain recovery at the time of the first post-discharge clinic visit.” [Page 20, Lines 385–391 of the Marked Copy]

Page 21, Line 401; Fig. 5;

This curve suggests that day 3 is a better indicator than days 1 and 2, and day 4 is a better indicator than day 3. Does this also suggest that day 5 (or day 6...) may be a better predictor than day 4?

[Authors’ response]

As mentioned in our response to your comment above, in terms of predictive power, the pain level on days 5, 6, and 7 seemed better than that on day 4 (AUC, 0.755, 0.842, and 0.823 vs. 0.723, respectively). However, we selected day 4 for further analysis as we identified day 4 as the earliest timepoint to predict pain recovery at the time of the first post-discharge clinic visit (Figure 5). This was based on the concept that earlier prediction of pain recovery would give us more chances for intervention to improve the patients’ outcomes. 

In response to the reviewer’s comment, we have revised the Methods section to clarify the reason why we focused on the pain level at the earliest timepoint to predict pain recovery as follows: 

“As a secondary analysis, the accuracy of pain severity on days 1, 2, 3, 4, 5, 6, and 7 for predicting recovery of pain to a mild level at the first post-discharge clinic visit was determined using area under the receiver operating characteristic (ROC) curves. In this analysis, the cut-off level of pain at the earliest timepoint to predict pain recovery at the time of the first post-discharge clinic visit was also determined by the ROC curve. This was based on the concept that earlier prediction of pain recovery would give us more chances for intervention to improve the patients’ outcomes.” [Pages 10-11, Lines 236-244 of the Marked Copy]

“Univariate analysis was used to evaluate the cut-off level of pain at the earliest timepoint determined by the ROC curve” [Page 11, Lines 245-247 of the Marked Copy]

Further, we have also revised Figure 5 and the Results section as follows:

“The ROC curves of pain severity level on days 1, 2, 3, 4, 5, 6, and 7 for predicting recovery of pain to mild level at the time of the first post-discharge clinic visit are shown in Fig 5. The area under the curve (AUC) of pain severity level on days 1, 2, 3, 4, 5, 6, and 7 were 0.502, 0.549, 0.626, 0.723, 0.755, 0.842, and 0.823 (P = 0.98, P = 0.48, P = 0.069, P = 0.001, P < 0.001, P < 0.001, and P < 0.001), respectively. Thus, day 4 was identified as the earliest timepoint to predict pain recovery at the time of the first post-discharge clinic visit.” [Page 19, Lines 385–391 of the Marked Copy]

Page 32, Line 507;

How do you think the absolute value of pain intensity affected this result? Did patients with stronger pain in the early postoperative period rebound more during the chronic period?

Does this association between pain on day 4 and chronic pain relate to so-called "pain memory"?

[Authors’ response]

 As shown in Figure 5, pain levels on days 4-7, not those on days 1-3, were associated with pain recovery to a mild level at the time of the first post-discharge clinic visit. This association may further support the concept of “pain memory,” where peripheral and central pain sensitization play a pertinent role in pain chronification.

In response to the reviewer’s comment, we have also revised the Discussion and References sections as follows:

 “Interestingly, chronic pain and memory share the anatomical sites of synaptic plasticity such as the limbic-cortical pathways [36]. Further, neuroimaging studies have suggested that the chronification of pain may result from persistence of pain memory via spatiotemporal reorganization of the neocortex [36].” [Pages 33–34, Lines 513–516 of the Marked Copy]

“This may indicate the possible association of delayed recovery of acute postsurgical pain with persistence of pain memory.” [Pages 34, Lines 518–520 of the Marked Copy]

[36] McCarberg B, Peppin J. Pain Pathways and Nervous System Plasticity: Learning and Memory in Pain. Pain Med. 2019;20: 2421-2437. doi: 10.1093/pm/pnz017. PMID: 30865778. [Page 42, Lines 707–709 of the Marked Copy]

Additionally, we recognized that the citation of Table numbers in the text required the following corrections:

(Table 3) [Page 28, Line 472 of the Marked Copy]

(Table 2) [Page 34, Line 518 of the Marked Copy]

We sincerely apologize for this confusion.

---

## [Decision Letter · Decision Letter 2]

7 Feb 2023

Symptom severity trajectories and distresses in patients undergoing video-assisted thoracoscopic lung resection from surgery to the first post-discharge clinic visit

PONE-D-22-03682R2

Dear Dr. Saito,

We’re pleased to inform you that your manuscript has been judged scientifically suitable for publication and will be formally accepted for publication once it meets all outstanding technical requirements.

Kind regards,

Silvia Fiorelli

Academic Editor

PLOS ONE

Additional Editor Comments (optional):

Congratulations to the authors and thanks to the reviewers for the suggestions provided which really helped improve the quality of the manuscript

Reviewers' comments:

Reviewer's Responses to Questions

**Comments to the Author**

1. If the authors have adequately addressed your comments raised in a previous round of review and you feel that this manuscript is now acceptable for publication, you may indicate that here to bypass the “Comments to the Author” section, enter your conflict of interest statement in the “Confidential to Editor” section, and submit your "Accept" recommendation.

Reviewer #3: All comments have been addressed

Reviewer #4: All comments have been addressed

2. Is the manuscript technically sound, and do the data support the conclusions?

Reviewer #3: Yes

Reviewer #4: Yes

3. Has the statistical analysis been performed appropriately and rigorously? 

Reviewer #3: Yes

Reviewer #4: Yes

4. Have the authors made all data underlying the findings in their manuscript fully available?

Reviewer #3: Yes

Reviewer #4: Yes

5. Is the manuscript presented in an intelligible fashion and written in standard English?

Reviewer #3: Yes

Reviewer #4: Yes

6. Review Comments to the Author

Reviewer #3: I asked several questions to the authors, all of which were answered appropriately.

This paper is considered suitable for acceptance.

Reviewer #4: The authors are presenting a study about post-operative symptoms/distress after VATS. I have no comments.

7. PLOS authors have the option to publish the peer review history of their article (what does this mean?). If published, this will include your full peer review and any attached files.

Reviewer #3: **Yes: **Yoshifumi Sano

Reviewer #4: No

---

## [Editor Report · Acceptance letter]

9 Feb 2023

PONE-D-22-03682R2 

Symptom severity trajectories and distresses in patients undergoing video-assisted thoracoscopic lung resection from surgery to the first post-discharge clinic visit 

Dear Dr. Saito:

I'm pleased to inform you that your manuscript has been deemed suitable for publication in PLOS ONE. Congratulations! Your manuscript is now with our production department. 

Kind regards, 

on behalf of

Dr. Silvia Fiorelli 

Academic Editor

PLOS ONE